# Synthesis of a Novel Spirocyclic Inflatable Flame Retardant and Its Application in Epoxy Composites

**DOI:** 10.3390/polym12112534

**Published:** 2020-10-29

**Authors:** Kunpeng Song, Yinjie Wang, Fang Ruan, Weiwei Yang, Jiping Liu

**Affiliations:** School of Materials Science and Engineering, Beijing Institute of Technology, 5 Zhongguancun South Street, Haidian District, Beijing 100081, China; 13439970074@163.com (K.S.); 18255171997@163.com (F.R.); 3120170543@bit.edu.cn (W.Y.)

**Keywords:** flame retardant, epoxy resin, spirocyclic, flame retardant mechanism

## Abstract

Derivatives of 3,9-dichloro-2,4,8,10-tetraoxa-3,9-diphosphaspiro-[5,5]undecane-3,9-dioxide (SPDPC) are of increasing interest as flame retardants for polymeric materials. In addition, SPDPC is also an important intermediate for the preparation of intumescent flame retardants (IFRs). However, low efficiency and undesirable dispersion are two major problems that seriously restrain the application of IFRs as appropriate flame retardants for polymer materials. Usually, the functionalization or modification of SPDPC is crucial to acquiring high-performance polymer composites. Here, a small molecule spirocyclic flame retardant diphenylimidazole spirocyclic pentaerythritol bisphosphonate (PIPC) was successfully prepared through the substitution reaction between previously synthesized intermediate SPDPC and 2-phenylimidazole (PIM). Phenyl group and imidazole group were uniformly anchored on the molecular structure of SPDPC. This kind of more uniform distribution of flame retardant groups within the epoxy matrix resulted in a synergistic flame retardant effect and enhanced the strength of char layers to the epoxy composites, when compared to the unmodified epoxy. The sample reached a limiting oxygen index (LOI) of 29.7% and passed with a V-0 rating in the UL 94 test with the incorporation of only 5 wt% of as-prepared flame retardant PIPC. Moreover, its peak of heat release rate (pHRR) and total heat release (THR) decreased by 41.15% and 21.64% in a cone calorimeter test, respectively. Furthermore, the addition of PIPC has only slightly impacted the mechanical properties of epoxy composites with a low loading.

## 1. Introduction

Epoxy resin is one of the three general-purpose thermosetting resins and is widely used in various fields, such as adhesives and coatings, electronics and electrical appliances, transportation and manufacturing, as well as aeronautics and astronautics [1,2,3,4,5]. However, both the highly flammability and smoke release of epoxy resins seriously restrain their application requirements of high-performance materials. So far, directly adding flame retardants into epoxy resins can effectively alleviate this situation [6,7].

Intumescent flame retardants (IFRs) are generally regarded as one of the most promising alternatives to traditional halogen flame retardants [8,9,10], and countries across the world have turned to halogen-free flame retardants to address environmental pollution. The intumescent flame retardant has the advantages of halogen-free, low smoke, low toxicity, anti-melting drip and low corrosive gas, etc., which is in line with the development direction of the future flame retardant industry and presents a broad application prospect [8,11,12]. Compared with halogen flame retardants, it has favorable flame retardant performance, accompanied by less toxic gas and smoke output, which has been confirmed by a large number of studies [13,14,15].

The use of a monocomponent IFR is a new strategy that includes three main parts of an IFR in a compound. This integrated method not only decreases the amount of flame retardant, but it also has a positive effect on the improvement of thermal stability and the dispersion between the parts [8,16]. SPDPC (3,9-dichloro-2,4,8,10-tetraoxa-3,9-diphosphaspiro-[5,5]undecane-3,9-dioxide) is a highly symmetrical cage-like compound that combines acid source and carbonization agent. It is able to promote the dehydration of oxygen-containing compounds to char during combustion. Moreover, SPDPC contains active P–Cl bonds at both ends, which is one of the ideal intermediates for preparing intumescent flame retardants.

In the past several years, spiral-containing compounds have proven to be a very effective flame retardants for polymer resins. Wang reported the preparation of poly(DOPO substituted dihydroxyl phenyl pentaerythritol diphosphonate) (PFR) via the polycondensation of SPDPC and 10-(2,5-dihydroxyl phenyl)-9,10-dihydro-9-oxa-10-phosphaphenanthrene-10-oxide (DOPO-BQ). The abundant aromatic structure and high phosphorus content of PFR provide excellent flame retardancy for epoxy composites without significantly reducing mechanical properties [17]. An intumescent flame retardant benzoguanamine spirocyclic pentaerythritol bisphosphonate (BSPB) was synthesized and used as a chain extender in the waterborne polyurethane (WPU) backbone, thus presenting higher maximum decomposition temperature and char residues [18]. Jiang synthesized imidazole spirocyclic phosphoramidate (ISPA) through a substitution reaction between SPDPC and imidazole, and used it to improve the flame retardancy of cotton fabrics. However, excessive additions and severe smoke release severely limit its application [19]. Nanocomposite technology has proposed a revolutionary new strategy for improving the fire resistance and mechanical properties of polymeric materials [20], such as layered double hydroxides (LDHs) [21,22], carbon-based material [23], metallic oxide [24], phyllosilicates [25], etc. Furthermore, modification of traditional inorganic hybrid nanomaterials often shows more significant effects [26]. Functionalized graphene nanosheets (GNSs) with covalently grafted polyphosphamide (PPA), i.e., PPA-gGNS, was incorporated into epoxy resins, exhibiting superior fire safety and mechanical strength. The improved performance was mainly attributed to the synergistic effect of PPA and graphene with a large aspect ratio [27]. Cai reported a functionalized layered double hydroxide (SPDP-LDH), by means of the coprecipitation reaction between spirocyclic pentaerythritol bisphosphorate disphosphoryl chlorine (SPDP) and Mg-Al-LDH (N-LDH) under hydrothermal conditions, aiming to inhibit agglomeration of N-LDH and improve the flame retardancy to epoxy composites with a low loading [28].

Following the idea above, in this study, a novel IFR diphenylimidazole spirocyclic pentaerythritol bisphosphonate (PIPC) was obtained through intermediate SPDPC and 2-phenylimidazole. The PIPC was incorporated into the epoxy resin system consisting of diglycidylether of bisphenol A (DGEBA) and 4,4-diaminodiphenylmethane (DDM) as a hardener to prepare epoxy composites. Compared with the traditional phosphorous flame retardants such as 9,10-dihydro-9-oxa-10-phosphaphenanthrene-10-oxide (DOPO) and 1-oxo-4-hydroxymethyl-2,6,7-trioxa-l-phosphabicyclo[2.2.2]octane (PEPA) [29], PIPC can achieve an ideal flame retardant effect with a relatively low additional amount. Such a SPDPC moiety contains a high phosphorus content, and phenylimidazole moiety comprises a high nitrogen content and rich aromatic structures, resulting in awesome thermostability and fire safety.

## 2. Experimental Section

### 2.1. Materials

Phosphorus oxychloride (≥98%), pentaerythritol (≥98%), and glacial acetic acid (98%) were purchased from the Tianjin Nankai Yungong Synthesis Technology Co., Ltd. (Tianjin, China), while 2-phenylimidazole (PIM) was provided by Guangzhou Jinsheng Chemical Co., Ltd. (Guangzhou, China). Tetrahydrofuran (AR), acetonitrile (AR), and 4,4-diaminodiphenylmethane (≥98%) were obtained from the Beijing Chemical Plant (Beijing, China). Acetone, dichloromethane, and triethylamine (TEA) were procured by Weiss Chemical Reagent Co., Ltd. (AR, Beijing, China). The diglycidyl ether of bisphenol A (E-44) in the study was purchased from Nantong Xingchen Synthetic Material Co., Ltd. (Hunan, China). All solvents in the study were dried over 4 Å molecular sieves before use. Pentaerythritol was ground and screened with 200 mesh copper screen before use.

### 2.2. Synthesis of PIPC

The PIPC was prepared by a two-step process, and the process is shown in Scheme 1. The synthesis method of intermediate SPDPC was improved according to the typical method reported by Liu [2]. Briefly, pentaerythritol (27.23 g; 0.2 mol) and acetonitrile (500 mL) were added to a 1 L three-neck flask and ultrasonically dispersed for 30 min at 40 °C. Then, we transferred it to an oil bath under nitrogen atmosphere, and the mixture was heated at 50 °C for 10 min in the stirring state. Subsequently, phosphorus oxychloride (122.66 g; 0.8 mol) was added to the above mixture all at once and stirred for 2 h at 60 °C. Immediately the reaction temperature was raised to 80 °C for 16 h and stirred for another 4 h with a reaction temperature up to 100 °C. After cooling down to room temperature, the mixture was filtered. In addition, the product was washed with acetone and dichloromethane to obtain white powder. This procedure was repeated for at least three times in order to remove the unreacted phosphorus oxychloride. After dried at 80 °C for 8 h, the white powder was recrystallized in glacial acetic acid to obtain a white powder SPDPC of higher purity. Lastly, the SPDPC was dried at 105 °C for 12 h in a yield of 73.8%.

The PIPC was prepared by a simple nucleophilic substitution reaction where 18.7 g 2-phenylimidazole was completely dissolved in 100 mL tetrahydrofuran by stirring under a nitrogen atmosphere. Subsequently, a few drops of TEA and SPDPC (14.9 g) were added to the above mixture. After stirring for 20 min at room temperature, the mixture was stirred for 1 h at 50 °C and then stirred for another 8 h with a reaction temperature up to 65 °C. After that, the mixture above was filtered, and the filter cake collected was washed 3 times with tetrahydrofuran and dichloromethane, successively. Lastly, the final white powder PIPC was vacuum dried at 70 °C for 12 h in a yield of 92.5%.

### 2.3. Preparation of the EP/PIPC Composites

A typical preparation of epoxy composites is illustrated in Scheme 2. Briefly, different mass fractions of PIPC (1, 2, 3, 4, and 5 wt%) were added to the epoxy resin with mechanical stirring for 1 h at 80 °C. Subsequently, the epoxy mixture was placed in the vacuum oven at 80 °C for 3 min to remove bubbles. After that, the epoxy mixture was maintained at 80 °C in an oil bath, followed by the addition of molten DDM (ratio of epoxy / DDM was 4:1) and stirring for 1 min to form a homogeneous liquid, and then it was instantly transferred into a vacuum oven at 80 °C for 1 min to remove bubbles again. Subsequently, the homogeneous liquid obtained was immediately poured into pre-heated poly (tetrafluoroethylene) molds of certain sizes. The curing temperature was set as 120 °C (2 h) and 150 °C (2 h). Eventually the prepared samples were cooled down naturally and denoted as EP/PIPC-1, EP/PIPC-2, EP/PIPC-3, EP/PIPC-4, and EP/PIPC-5. Similar to the manufacturing process of the EP/PIPC samples, the neat EP sample was prepared without the flame retardant additive PIPC.

### 2.4. Characterization

A Tensor 27 IR spectrometer (Bruker Optics, Beijing, China) was applied to conduct a Fourier-transform infrared (FTIR) spectra test.

Nuclear magnetic resonance (NMR) spectroscopy was also conducted: ^1^H NMR and ^31^P NMR spectra were performed on an FT-80A NMR spectrometer (Varian, CA, USA), using deuterated dimethyl sulfoxide (DMSO-d_6_) as the solvent.

A thermogravimetric analysis (TGA) and differential scanning calorimetry (DSC) were applied with the Toledo STARe thermogravimetric analyzer (Mettler-Toledo, Zurich, Switzerland). Measurements were carried out from 50 to 800 °C at a heating rate of 20 °C/min and from 50 to 180 °C with a heating rate of 5 °C/min under a nitrogen atmosphere.

Limiting oxygen index (LOI) tests were employed using an FTAII 1600 LOI instrument (Phoenix Instruments Co., Ltd., Suzhou, China), and according to the standard ASTM D 2863 procedure on samples with 130 × 6.5 × 3 mm^3^.

Vertical burning tests were utilized with a CZF-5 horizontal vertical burning tester (Phoenix Instruments Co., Ltd., Suzhou, China) using the UL 94 standard. The sample dimensions were 130 × 13 × 3 mm^3^.

Cone calorimeter measurements were employed using a Fire Testing Technology (FTT) apparatus (Phoenix Instruments Co., Ltd., Suzhou, China), and the measurements carried out according to the ISO 5660 protocol, accompanied by an incident radiant flux of 50 kW/m^2^. The sample dimensions were 100 × 100 × 3 mm^3^.

Scanning electron microscopy (SEM) were applied with a Hitachi SU8020 (Hitachi Limited, Tokyo, Japan), accompanied by an accelerating voltage of 15 kV.

X-ray photoelectron spectroscopy (XPS) patterns of the samples were performed by a Quantera II X-ray photoelectron spectroscopy (Ulvac-PHI, Chigasaki, Japan).

We used the Toledo STARe thermogravimetric analyzer (Mettler-Toledo, Zurich, Switzerland) coupled with Fourier-transform infrared spectroscopy (Bruker Optics, Beijing, China) through a transfer pipe to conduct the thermogravimetric analysis/infrared spectrometry (TG-IR) test from 50 to 800 °C at a heating rate of 20 °C/min^−1^ under nitrogen atmosphere.

Mechanical properties were tested by employing an electronic tensile testing machine (DXLL-5000, Shanghai, China) with a rate of 2 mm/min.

## 3. Results

### 3.1. Characterization of the Structure of and Thermal Stability PIPC

As shown in Figure 1, the typical peaks of the SPDPC spectrum at 1320, 1150, and 551 cm^−1^ correspond to P=O, P–O–C, and P–Cl stretching vibrations; the absorption peak at 2983 and 1468 cm^−1^ are attributed to –CH_2_– in the spirocyclic moiety stretching and bending vibrations, respectively [30,31,32]. For the PIPC spectrum, there is no obvious P–Cl absorption peak at 551 cm^−1^. However, a new absorption peak appears at 1635 cm^−1^, associated with the C=C characteristic absorption of the PIM. PIM presents a distinct bending vibration peak corresponding to the N–H bond at 1562 cm^−1^, but PIPC does not have any absorption peaks here. Based on these analysis above, it is preliminarily proven that PIPC was successfully synthesized.

To further demonstrate the chemical structure of the target product, ^1^H NMR and ^31^P NMR tests were obtained (Figure 2). As reported in the previous literature [2,32], Figure 2a,b presents standard ^1^H NMR and ^31^P NMR spectra of intermediate SPDPC. Two contour characteristic peaks at around 4.21 and 4.25 ppm are resonance peaks of the methylene hydrogen atoms attached in a spiral ring. Moreover, the only one single peak at around −7.2 ppm corresponds to the characteristic peak of phosphorus. The molecular structure of PIPC is shown in Figure 2c,d. Its ^1^H NMR spectrum retains methylene characteristic peaks of the spirocyclic moiety, with emerging aromatic hydrogen signals at wide chemical shift ranges. Additionally, its ^31^P NMR spectrum is quite similar to SPDPC, showing the phosphorus atoms to be in the same chemical environment.

The thermal stability of PIPC was monitored under nitrogen atmosphere (Figure 3). PIPC shows a one-stage thermal decomposition process, mainly attributed to the degradation of the spirocyclic moiety and aromatic group. For PIPC, the initial decomposition temperature (*T*_5%_) is 278.6 °C, and the maximum degradation temperatures (*T*_max_) are 316.6 and 397.0 °C with approximately 42.1% residual char at 800 °C, showing that its thermal stability is significantly better than SPDPC (Table 1). This is possibly attributed to the introduction of aromatic structures and nitrogen-containing groups that facilitates the compounds’ decomposition to form residues with a higher degree of crosslinking. The flame retardant PIPC presents good thermal stability and high residual char, which could meet the processing requirements of epoxy resin.

### 3.2. Analysis on Morphology and Dispersion of EP/PIPC Composites

SEM-EDS was applied to analyze the cross-sectional morphology and microdomain element distribution of the EP control and EP/PIPC-5 (Figure 4). From a macro perspective, although prepared epoxy composites are slightly darker than the EP control, they all show better transparency, which also proves that PIPC has good dispersibility in epoxy resins (Figure 4a). We can observe that PIPC is more uniformly dispersed in the epoxy resin without obvious particles (Figure 4c). In addition, the P and N elements in the EP/PIPC-5 exhibit uniform dispersion (Figure 4d,e).

### 3.3. Analysis on Thermal Properties of EP/PIPC Composites

From Figure 5a and Table 2, the initial decomposition temperature of EP/PIPC composites presents a gradual downward trend with the increasing PIPC loading. EP/PIPC composites are less thermally stable than the EP control when evaluated by *T*_5%_ and *T*_max_; this is possibly attributed to the catalysis degradation initiated by PIPC thermal degradation in advance. Interestingly, the incorporation of PIPC has little effect on the maximum decomposition temperature of EP/PIPC composites. As the content of PIPC in epoxy composites increases, glass transition temperature (*T*_g_) shows a downward trend. It may be attributed to the presence of PIPC molecules in the epoxy composite, which increases the distance between the epoxy group in the epoxy monomer and the amino group in the hardener, resulting in a decrease in the degree of crosslinking of the epoxy resin matrix. As the amount of PIPC increases, the crosslinking degree of epoxy composite decreases more obviously, making the *T*_g_ show a downward trend (Figure 5c). As far as the residual char yield is concerned, the incorporation of PIPC results in the improvement of the char residues at 600 °C and 800 °C, which is due to the catalytic carbonization effect of PIPC during decomposition. Taken together, the thermal analysis results indicate that the aromatic structure and spirocyclic moiety jointly contribute to improving the thermal performance of epoxy composites.

### 3.4. Analysis on Fire Safety of EP/PIPC Composites

The LOI and UL 94 levels of the EP control and EP/PIPC composites are displayed in Figure 5d. The LOI values of all the samples were gradually enhanced with the increasing of the content of PIPC. When the PIPC content is 5 wt%, the LOI value of the epoxy composite rises to 29.7%, achieving a UL 94 V-0 rating without dropping. It is worth noting that the additive with a reduced PIPC dosage of 3 wt% still achieved V-1 rating. The above analysis shows that PIPC can effectively enhance the flame retardancy of the epoxy matrix.

Cone calorimetry testing was performed on those samples to further evaluate the detailed combustion behavior. The heat release rate (HRR), total heat release (THR), and total smoke production (TSP) curves with the associated data of epoxy composites are shown in Figure 6 and Table 3. Compared to the EP control, the addition of 5 wt% PIPC brought about a 41.15% maximum decrease in the peak of heat release rate (pHRR), a 23.69% maximum decrease in THR (at 200s), and a 12.5% maximum decrease in TSP. The pHRR of epoxy composites decreased from 1023 to 602 kW/m^2^ after adding 5 wt% PIPC, indicating that the fire hazard of epoxy composites could be significantly reduced in real fire situations. In addition, the HRR curves of EP/PIPC-5 presented two peaks, which is ascribed to the breaking of the lower-strength char layer formed earlier. Compared with the EP control, the pHRR of EP/PIPC-5 showed a significant delay. This also shows from the side that the introduction of PIPC could promote the decomposition and carbonization of epoxy resin, which is beneficial to the formation of high-strength char layers. Moreover, the formation of dense char layer exerts a good barrier to the flammable gas and smoke particles [33], so that EP/PIPC composites displayed a lower THR and TSP value as compared to the EP control.

As the content of flame retardants increases, the CO_2_ release rate decreased significantly (Figure 6d). The slight decrease of CO_2_ formation indicates that phosphorus-containing radicals may be of significance. Moreover, the CO release rate also decreased slightly, which may be inseparable from some carbon-containing compounds being transferred to the condensed phase (Figure 6e).

We know that phosphorus-containing flame retardants can significantly enhance the char formation of resin containing oxygen. However, they are not effective for lowering smoke emissions [34,35]. The incorporation of PIPC did not significantly inhibit the TSR of epoxy resin, indicating that the gas-phase flame retardant mechanism in the EP/PIPC composites played a significant role.

### 3.5. Analysis on the Flame-Retardant Mechanism of Condensed Phase for EP/PIPC Composites

The residual chars of the EP-control and EP composites after cone calorimeter tests were compared from different visual angles by digital photos (Figure 7a–d). The EP control exhibit porous and brittle char residue structure. It can be clearly observed that the residual char production of EP/PIPC-5 composites is higher than that of the EP control. This clearly shows that the introduction of PIPC has a significant effect on the char formation performance in the epoxy combustion process.

Figure 7e–h shows the SEM morphology of the EP control and EP/PIPC-5 char residue. It can be seen that the interior and exterior char layers of the EP control were severely broken, and large irregular holes were formed, mainly due to the fragile char layer being easily damaged by the gas generated during decomposition of the substrate. Interestingly, the interior char layer of EP/PIPC-5 formed three different types of “bubbles”, namely, fully broken bubbles, semi-ruptured bubbles and unruptured bubbles. This is mainly ascribed to the good strengthening effect of PIPC on the strength of the char layer. Compared with the EP control, the exterior char layer of EP/PIPC-5 appeared continuous and smooth, forming a raised “airbag” structure. These “bubble” structures exhibited good channel and barrier effects, effectively slowing the heat release rate [36,37]. Notably, this is the main reason why EP/PIPC-5 presents better fire safety.

XPS characterizations were employed to further study the interior and exterior char residue of the EP control and EP/PIPC-5, and the results are shown in Figure 8. The internal and external char residue of EP/PIPC-5 present the characteristic peaks of phosphorus, and the phosphorus content in the external residual char is higher than that of the internal residual char, indicating that the phosphorus-containing compounds generated by combustion tend to migrate to the external. In addition, compared with the EP control, the content of O element in the external char residue of EP/PIPC-5 increased. This is mainly due to the combination of O and P elements to form phosphorus-containing compounds such as O_2_P^+^, OP^+^ and HO_2_P^+^, which also makes the char layer presents better thermal oxidation stability [38,39].

To further study the degradation process of the condensed phase for EP/PIPC-5, FTIR was utilized to test the residual coke after the segmented cone calorimetry test (Scheme 3) [40]. Char residue formed by the EP control and EP/PIPC-5 at 50, 100, 150, 200, 250, and 300 s were characterized in Figure 9. For the EP control, the internal and external char layers still maintained the original structure within 50 s. After 50 s, the characteristic peaks at 2919 and 2858 cm^−1^ corresponding to fatty chains were significantly reduced, and the typical characteristic peaks at 1600 to 1650, 1380, and 1395 cm^−1^ that were attributed to aromatic structures showed the same trend. The char layer structure showed significant changes, which was mainly attributed to the breakdown of the crosslinked structure inside the epoxy resin. Subsequently, a broad absorption peak appeared at about 1050 cm^−1^, and the absorption peak of the external residual char appeared wider than that of the internal. This indicates that the residual char was oxidized to from a new C–O structure, while the external char layer was more oxidized than that of the internal one during the combustion process. Compared with the EP control, at 50 s, the internal and external char layer structure of EP/PIPC-5 changed, which is mainly caused by the PIPC promoting early decomposition of epoxy resin. In addition, the internal and external residual chars of EP/PIPC-5 tend to stabilize after 200 s, while that of the EP control still changes. This is mainly attributed to the PIPC promoting the dehydration of the epoxy substrate to charcoal and forming char layers with good stability in advance.

### 3.6. Analysis of Gas Phase Flame Retardant Mechanism of EP/PIPC Composites

The gas phase degradation process of the EP control and EP/PIPC-5 were studied with the TG-FTIR technique (Figure 10).

FTIR spectra of the pyrolysis products for the EP control and EP/PIPC-5 at different temperatures (350, *T*_max_, 500, 600 °C) are shown in Figure 11 and Table 4. On the whole, the addition of PIPC had no significant effect on the types of pyrolysis gas products of epoxy composites. However, the introduction of PIPC significantly changed the appearance time of gas products. At 350 °C, the EP control did not show obvious decomposition, while EP/PIPC-5 began to decompose the main gas products, which is attributed to PIPC’s lower initial decomposition temperature to promote the decomposition of the epoxy matrix in advance. At T_max_, the EP control began to decompose violently, and the absorption peaks corresponding to aromatic compounds, esters, and ethers were very intense. However, for EP/PIPC-5, the intensity of the absorption peaks at 3735 and 1744 cm^−1^ significantly weakened and almost disappeared, while the absorption peaks at other positions were significantly enhanced. It shows that free water vapor and carbonyl compounds disappear, while other gas products had a larger output at 350 °C to *T*_max_. At 500 °C, the infrared absorption peaks of the EP control and EP/PIPC-5 gas products were very similar, and there is no obvious difference. However, as the temperature increased, the absorption peaks at 3735, 3650, 1257, and 1178 cm^−1^ all weakened significantly, and the peak intensity at 3540–3340 cm^−1^ dramatically increased. This implies that the production amount of hydroxyl compounds, esters, or ether compounds began to gradually decrease, while the production amount of amine compounds increased significantly in *T*_max_ to 500 °C; In the end, the infrared absorption peaks of the two samples were almost unchanged, indicating that the decomposition of the flame retardant epoxy resin entered a stable stage at 500–600 °C.

The thermal degradation process of the EP control and EP/PIPC-5 composites was studied by TG-FTIR at different times (Figure 12). Comparing the decomposition process of the EP control and EP/PIPC-5, the positions of the absorption peaks in the spectra were basically the same, which indicates that the addition of PIPC does not change the kind of gas generated by the decomposition of the epoxy matrix. At the same time, it was confirmed from the side that most of the phosphorus products produced by the decomposition of PIPC remained in the condensed phase. It is worth noting that the absorption peaks of each gas product produced by EP/PIPC-5 decomposition were significantly weaker than that of the EP control, which may be due to the fact that PIPC can effectively suppress the amount of gas produced by epoxy decomposition. In addition, we can find that there is no obvious decomposition of the EP composites before 15 min. Until about 20 min, the gas products generated can be observed. When it reaches about 25 min, the gas production achieves the maximum, and then the gas production decreases and stabilizes.

To compare the evolution of gaseous products, the absorbance intensities and generation time of the representative pyrolysis products for the EP control and EP/PIPC-5 are presented in Figure 13. With the incorporation of 5 wt% PIPC, the maximum absorbance intensity of pyrolysis products were diverted to lower values, including ester/ether compounds, aromatic compounds, carbonyl compounds, hydrocarbons, and methane, compared with those of the EP control. Interestingly, the EP control and EP/PIPC-5 were basically the same as the main gas phase pyrolysis products, but the appearance time of the gas products of EP/PIPC-5 was earlier than that of the EP control, which is inseparable from PIPC’s promotion of epoxy resin decomposition in advance. The statistical analysis shows that the total release of ester/ether compounds, aromatic compounds, carbonyl compounds, hydrocarbons, and methane decreased by 43.65%, 32.36%, 46.32%, 21.68%, and 29.66%, respectively, indicating that the addition of PIPC greatly reduced the release of flammable gases (Figure 13h). In addition, the total release of ammonia and water vapor increased by 8.31% and 5.35%, respectively, which dilutes the combustible gas and oxygen concentration. It also proves that PIPC decomposition promotes epoxy resin dehydration and achieves the inhibiting combustion.

### 3.7. Analysis of Flame Retardant Mechanism of EP/PIPC Composites

Based on the above condensed phase and gas phase analysis, a possible flame retardant mechanism is proposed and explained in Scheme 4. It is clear from Figure 12 that the combustion of epoxy polymers produces many pyrolysis gases, including nitrogen hydrides, carbonyl compounds, aromatic compounds, hydrocarbons, and carbon oxides, which are considered to have smoke poisoning hazard performance. Due to the PIPC’s catalytic action and free radical trapping effect, the carbon oxides, hydrocarbons, and other pyrolytic gases generated during the EP composites decomposition are converted from the gas phase to the condensed phase, resulting in a highly graphitized char residue [41]. Generally speaking, PIPC acts as an IFR with a carbonization agent, acid source, and blowing agent. During the decomposition of the epoxy matrix, the acid source promotes the dehydration and carbonization of epoxy resin molecular chain and carbonization agent to form molten char layers, which become bulk with the gas generated by the blowing agent [12,42]. The char layers covering the surface of the matrix inhibit the heat and flammable gas exchange to a certain extent, thereby enhancing the thermal stability of the epoxy composites [43]. Furthermore, besides the role of PIPC in trapping free radicals in the gas phase, the crosslinked phosphorus oxynitride and carbonized aromatic network generated during the combustion process can also promote the formation of char residue. Therefore, the role of PIPC inhibits the emission of toxic gases and facilitates the formation of char layers, which effectively reduces HRR and THR, indicating that the fire safety of EP/PIPC composites is greatly improved.

### 3.8. Mechanical Properties of EP/PIPC Composites

The mechanical properties of epoxy composites were characterized by tensile and flexural tests. Figure 14 presents the tensile strength and flexural strength values related to PIPC content for epoxy composites. The tensile strength increased by 3.33% and 8.33% with 1 and 3 wt% loading of PIPC, respectively, compared with the EP control. However, the incorporation of 5 wt% PIPC resulted in a 6.67% decrease in tensile strength. The flexural strength of epoxy composites presented a similar trend to tensile strength with the increasing PIPC concentration. This is inseparable from the good dispersibility of PIPC in the epoxy matrix.

## 4. Conclusions

In present work, a small molecule spiro-ring flame retardant PIPC was designed and synthesized to endow epoxy composites with better fire safety performance. The LOI value of the EP/PIPC-5 increased from 25.1% to 29.7% as compared to that of the EP control, and passed the UL 94 test with a V-0 rating. In particular, compared to the EP control, the pHRR and THR of EP/PIPC-5 were reduced by 42.15% and 21.64%, respectively. The advantages of PIPC for the flame retardant included enhanced strength and oxidation stability of the char, suppressed emission of poisonous gases, and synergistic flame-retardant effect. In the condensed phase, PIPC decomposes in advance to produce phosphoric acid or acid anhydride, which promotes the dehydration and carbonization of the epoxy matrix, forming a continuous and compact char layer as a physical barrier. In the gas phase, besides the free radical trapping effect of PIPC, part of the gas produced by the decomposition of EP/PIPC composites accumulates in the char layer, forming a raised “balloon”. When the amount of gas accumulates to a certain extent, it will break through the char layer, spray out, and blow the flame away from the epoxy matrix. The above analysis shows that PIPC is an excellent, efficient, and potentially valuable epoxy resin additive.

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
