# Peer review of "Synthesis of a Novel Spirocyclic Inflatable Flame Retardant and Its Application in Epoxy Composites"

_polymers, 2020, doi:10.3390/polym12112534_

Round 1
Reviewer 1 Report
Derivatives of 3,9–dichloro–2,4,8,10–tetraoxa–3,9–diphosphaspiro–[5,5]undecane–3,9–dioxide (SPDPC) are of increasing interest as flame retardants for polymeric materials.
The authors of the presented study synthesized a flame retardant additive by the reaction of SPDPC with 2-phenylimidazole. That additive (denoted as PIPC within the manuscript) was incorporated into an epoxy resin system consisting of diglycidylether of bisphenol A (DGEBA) and 4,4-diaminodiphenylmethane (DDM) as hardener.
The synthesis of PIPC from SPDPC and 2-phenylimidazole have already be mentioned in a patent previously filed by authors of the study.
The novel additive showed good flame-retardant effect in the epoxy resin investigated. The best classification (V0) in the UL94-V-test was achieved at relatively low loading (5 wt%). In addition, investigations by cone calorimetry revealed that PIPC decreases the peak of heat release rate (pHRR) by about 40% reduceing the fire risk significantly. However, the following statement at page 7 (lines 225, 226) is not correct: “Compared with EP control, the pHRR of EP/PIPC–5 was significantly earlier …”. In fact, two peaks appear in the HRR curve but the second one is higher and therefore to assign to pHRR. That means pHRR of EP/PIPC–5 is reached later compared to the sample of neat resin.
The mode of action of PIPC was investigated thoroughly (char morphology, FTIR spectra of the pyrolysis products …). These investigations clearly revealed pronounced condensed-phase action of the novel flame retardant (intumescence, better char morphology). In my opinion, the relevance of gas phase action of PIPC could not fully clarified. The flame retardant causes a blow-out effect, but the statement “This is mainly due to the decomposition of PIPC to generate phosphorus–containing free radicals, which could quench H·, HO· and O· in the process of supporting the combustion chain reaction” is not well-grounded (only a slight decrease of CO2 formation indicates that phosphorus-containing radicals may of significance).
On the whole the manuscript is well written, provides relevant results and therefore it should be published in the journal polymers.
Author Response
Response to Reviewer 1 Comments
Responds to the reviewer’s comments:
For reviewer #1:
Thank you very much for your useful comments and suggestions on our manuscript, which help us a lot to improve our paper. We have made careful revisions on the original manuscript according to the comments. We hope the revised edition will meet the journal’s standard for publication.
Point 1: Derivatives of 3,9–dichloro–2,4,8,10–tetraoxa–3,9–diphosphaspiro–[5,5]undecane–3,9–dioxide (SPDPC) are of increasing interest as flame retardants for polymeric materials.
Response 1: Thank you for your careful reading and put it in a better way for our manuscript. In page 1 line 11, the sentence of “3,9–dichloro–2,4,8,10–tetraoxa–3,9–diphosphaspiro–[5,5]undecane–3,9–dioxide (SPDPC) was regarded as an outstanding flame–retardant additive for polymers.” was corrected as “Derivatives of 3,9–dichloro–2,4,8,10–tetraoxa–3,9–diphosphaspiro–[5,5]undecane–3,9–dioxide (SPDPC) are of increasing interest as flame retardants for polymeric materials.”. In addition, the revised sentence has been marked in the manuscript.
Point 2: The authors of the presented study synthesized a flame retardant additive by the reaction of SPDPC with 2-phenylimidazole. That additive (denoted as PIPC within the manuscript) was incorporated into an epoxy resin system consisting of diglycidylether of bisphenol A (DGEBA) and 4,4-diaminodiphenylmethane (DDM) as hardener.
Response 2: Thank you for your valuable advice. In page 2 line 75, the sentence of “a novel IFR diphenylimidazole spirocyclic pentaerythritol bisphosphonate (PIPC) was obtained through intermediate SPDPC and 2–phenylimidazole to prepare epoxy composites.” was corrected as “a novel IFR diphenylimidazole spirocyclic pentaerythritol bisphosphonate (PIPC) was obtained through intermediate SPDPC and 2–phenylimidazole. The PIPC was incorporated into the epoxy resin system consisting of diglycidylether of bisphenol A (DGEBA) and 4,4-diaminodiphenylmethane (DDM) as hardener to prepare epoxy composites.”.
Moreover, we added that compared with traditional phosphorus flame retardants (Ind. Eng. Chem. Res. 2017, 56, 5, 1245–1255), the PIPC in this study achieves a good flame retardant effect at a lower addition amount. In page 2 line 79, the sentence of “Compared with the traditional phosphorous flame retardants 9,10-dihydro-9-oxa-10-phosphaphenanthrene-10-oxide (DOPO) , 1-oxo-4-hydroxymethyl-2,6,7-trioxa-l-phosphabicyclo[2.2.2]octane (PEPA), etc., PIPC can achieve an ideal flame retardant effect with a relatively low addition amount.” was added. In addition, the revised sentence has been marked in the manuscript.
Point 3: The synthesis of PIPC from SPDPC and 2-phenylimidazole have already be mentioned in a patent previously filed by authors of the study.
Response 3: The patent of PIPC preparation for epoxy resin flame retardancy has been reported, which comes from the research results of this research group. Affected by coVID-19 at that time, the school could not carry out experiments, and some data were not yet complete. Therefore, the follow-up work was not completed until the epidemic was under control.
Point 4: The novel additive showed good flame-retardant effect in the epoxy resin investigated. The best classification (V0) in the UL94-V-test was achieved at relatively low loading (5 wt%). In addition, investigations by cone calorimetry revealed that PIPC decreases the peak of heat release rate (pHRR) by about 40% reduceing the fire risk significantly. However, the following statement at page 7 (lines 225, 226) is not correct: “Compared with EP control, the pHRR of EP/PIPC–5 was significantly earlier …”. In fact, two peaks appear in the HRR curve but the second one is higher and therefore to assign to pHRR. That means pHRR of EP/PIPC–5 is reached later compared to the sample of neat resin.
Response 4: Thank you for your valuable and thoughtful comments. Your guidance gave me a better understanding of HRR analysis. We are sorry for the lack of rigour in our interpretation. In page 8 line 246, the sentence of “Compared with EP control, the pHRR of EP/PIPC–5 was significantly earlier, which also shows from the side that the introduction of PIPC could promote the early decomposition and carbonization of epoxy resin, and protect the internal substrate” was corrected as “Compared with the EP control, the pHRR of EP/PIPC-5 showed a significant delay. This also shows from the side that the introduction of PIPC could promote the decomposition and carbonization of epoxy resin, which is beneficial to the formation of high-strength char layers.”. In addition, the revised sentence has been marked in the manuscript.
Point 5: The mode of action of PIPC was investigated thoroughly (char morphology, FTIR spectra of the pyrolysis products …). These investigations clearly revealed pronounced condensed-phase action of the novel flame retardant (intumescence, better char morphology). In my opinion, the relevance of gas phase action of PIPC could not fully clarified. The flame retardant causes a blow-out effect, but the statement “This is mainly due to the decomposition of PIPC to generate phosphorus–containing free radicals, which could quench H·, HO· and O· in the process of supporting the combustion chain reaction” is not well-grounded (only a slight decrease of CO2 formation indicates that phosphorus-containing radicals may of significance).
Response 5: According to your helpful advice. We are sorry for the lack of rigour in our interpretation. In page 9 line 259, the sentence of “This is mainly due to the decomposition of PIPC to generate phosphorus–containing free radicals, which could quench H•, HO• and O• in the process of supporting the combustion chain reaction” was deleted. Just like your point of view, the relevance of gas phase action of PIPC could not fully clarified in the manuscript.
What’s more, in page 9 line 258, “The slight decrease of CO2 formation indicates that phosphorus-containing radicals may of significance” was added. The revised sentence has been marked in the manuscript.
Moreover, we have made appropriate amendments to some references in the manuscript and amended as follows:
Replace “[1] Zhang, Z.D.; Qin, J.Y.; Zhang, W.C.; Pan, Y.T.; Wang, D.Y.; Yang, R.J. Synthesis of a novel dual layered double hydroxide hybrid nanomaterial and its application in epoxy nanocomposites. Chem. Eng. J. 2020, 381, 122777.” with “[1] Zotti, A.; Borriello, A.; Zarrelli, M.; Zuppolini S. Thermal Properties and Fracture Toughness of Epoxy Nanocomposites Loaded with Hyperbranched-Polymers-Based Core/Shell Nanoparticles. Nanomaterials. 2019, 9(3), 418.”;
Replace “[3] Jiang, S.D.; Tang, G.; Chen, J.; Huang, Z.Q.; Hu, Y. Biobased polyelectrolyte multilayer-coated hollow mesoporous silica as a green flame retardant for epoxy resin. J. Hazard. Mater. 2018, 342, 689-697.” with “[3] Gupta, P.; Bajpai, M. Development of Siliconized Epoxy Resins and Their Application as Anticorrosive Coatings. Advances in Chemical Engineering & Science. 2011, 1(3):133-139.”;
Replace “[6] Yu, B.; Xing, W.; Guo, W.; Qiu, S.L.; Wang, X.; Lo, S.M.; Hu, Y. Thermal exfoliation of hexagonal boron nitride for effective enhancements on thermal stability, flame retardancy and smoke suppression of epoxy resin nanocomposites via sol-gel process. J. Mater. Chem. A. 2016, 4, 7330-7340.” with “[6] Gérard, C.; Fontaine, G.; Bourbigot, S. New Trends in Reaction and Resistance to Fire of Fire-retardant Epoxies. Materials. 2010, 3(8):4476-4499.”;
Replace “[9] Chen, M.J.; Shao, Z.B.; Wang, X.L.; Chen, L.; Wang, Y.Z. Halogen-Free Flame-Retardant Flexible Polyurethane Foam with a Novel Nitrogen-Phosphorus Flame Retardant. Ind. Eng. Chem. Res. 2012, 51(29), 9769-9776.” with “[9] Vladimir, Y.; Arnis, A.; Dzintra, V.; Irina, S. Polyurethane Coatings Based on Linseed Oil Phosphate Ester Polyols with Intumescent Flame Retardants. Fire & Materials. 2019, 43(1), 92-100.”;
Replace “[11] Chen, H.D.; Wang, J.H.; Ni, A.Q.; Ding, A.X.; Han, X.; Sun, Z.H. The effects of a macromolecular charring agent with gas phase and condense phase synergistic flame retardant capability on the properties of PP/IFR composites. Materials 2018, 11(1), 111.” with “[11] Saheb, S.; Tambe, P.; Malathi, M. Influence of Halloysite Nanotubes and Intumescent Flame Retardant on Mechanical and Thermal Properties of 80/20 (wt/wt) PP/ABS Blend and Their Composites in The Presence of Dual Compatibilizer. Journal of Thermoplastic Composite Materials. 2018, 31(2), 202-222.”.
Your Suggestions are of great significance to my future work. Special thanks to you for your good comments.

Reviewer 2 Report
This manuscript, polymes-961058, reports interestig results of experimental sudies on the effects of a synthesized flame retardant, PIPC, on epoxy resin. Improvements in the following aspects are recommended before the publication in polymers.
1) Dissolution or dispersion of flame retardants in polymer systems is one of important properties for the appropriate applications. Solubility of PIPC in the epoxy resin should be described in the manuscript.
2) DSC thermograms are given Gigure 4 (c). But, details of the DSC and the measurements were not given in the text. Why Tg's of the cured epoxy resins containing PIPC decrese with increasing PIPC contents?
3) What is the temperture for the FT-IR measurements shown in Figure 11?
4) Improvements of flame retardancy by PIPC should be mentioned in comparison with one of conventional flame retardants.
Round 2
Reviewer 2 Report
Moderate English changes are required especially for the revised (highlighted) texts before the publication.